# Artificial intelligence-enabled electrocardiography from scientific research to clinical application

Chin-Sheng Lin [ID] [1,2,3], Wei-Ting Liu [ID] [1], Yuan-Hao Chen [4], Shih-Hua Lin [5] & Chin Lin [ID] [2,3] [✉]

## Abstract

Recent advancements in artificial intelligence (AI) have revolutionized the application of electrocardiography (ECG) in cardiovascular diagnostics. This review highlights the transformative impact of AI on traditional ECG analysis, detailing how deep learning algorithms are overcoming the limitations of human interpretation and conventional diagnostic criteria. Historically, ECG interpretation has relied on well-established, physiologically-based criteria. The advancement of AI-ECG is marked by its capacity to process complex high-dimensional data directly from raw signals, revealing patterns often missed by conventional methods. Notably, AI models have identified signs of asymptomatic low ejection fraction and paroxysmal atrial fibrillation during normal sinus rhythm, enabling earlier clinical intervention. In addition to improved diagnostic utility, AI-ECG offers promising applications in risk stratification and community screening. Several randomized controlled trials (RCTs) have shown that integrating AI into clinical workflows not only reduces critical intervention times but also identifies patients at elevated risk of adverse outcomes. Future directions involve integrating additional clinical data sources, improving model interpretability through explainable AI, and developing unified platforms to manage outputs from multiple models.

**Keywords** Artificial Intelligence; Electrocardiography; Digital Biomarker; Paradigm Shift; Opportunistic Screening
**Subject Categories** Biotechnology & Synthetic Biology; Cardiovascular System; Computational Biology

## Introduction

In recent years, artificial intelligence (AI) has significantly transformed various domains (Wang et al, 2023), with healthcare being among the most affected (Rajpurkar et al, 2022). Recent studies report that each year in the United States, 800,000 people suffer severe disability or death due to diagnostic errors (Newman-Toker et al, 2024). Many diseases, such as heart failure and sudden cardiac arrest, are diagnosed late owing to subtle signs or differences in clinical judgment. In addition to detecting patterns earlier, medical AI models can meticulously evaluate patients' test results and medical records, closely mirroring physician expertise, and deliver precise diagnostic assessments and tailored treatment recommendations. Consequently, AI models are regarded as one of the most promising tools for addressing diagnostic errors (Topol, 2024). Moreover, physicians are confronted with overwhelming data volumes and administrative burdens. AI-driven automation and actionable insights can alleviate cognitive load and free up time for more focused patient-centered care. Accordingly, the integration of AI into healthcare settings has opened new avenues for diagnostics, predictive analytics, patient management, and personalized medicine.

The adoption of AI in healthcare is primarily driven by the increasing availability of health data and advancements in machine learning and deep learning technologies (Goldstein et al, 2017). Electronic health records (EHRs), imaging data, genomics, and real-time monitoring from wearable devices contribute to a data-rich environment that can be used by AI algorithms. Unlike traditional statistical models, AI systems, particularly those powered by deep learning, can analyze complex, high-dimensional data patterns (LeCun et al, 2015). As a result, these systems can support healthcare professionals by providing high-accuracy predictions, often surpassing human performance in tasks such as image interpretation and early disease detection. In particular, diagnostic AI models have proven to be effective in interpreting radiology images, predicting disease progression, and supporting timely interventions (Rajpurkar and Lungren, 2023). These models help mitigate diagnostic errors and ensure that patients receive appropriate care faster, addressing a critical challenge in modern healthcare.

Among AI applications in diagnostics, 12-lead resting electrocardiography (ECG) has gained particular interest due to its central role in cardiovascular health management (Friedman, 2024). ECG is a widely used non-invasive tool that records the heart's electrical activity and provides essential information for diagnosing various cardiac conditions, including arrhythmias, myocardial ischemia and injury, and conduction disturbances. Given its simplicity, accessibility, and low cost, ECG is utilized globally in both emergency and routine healthcare settings. However, the accurate interpretation of ECG data demands considerable expertise, and

[1]Division of Cardiology, Department of Internal Medicine, Tri-Service General Hospital, National Defense Medical University, Taipei, Taiwan, ROC. [2]Medical Technology Education Center, School of Medicine, College of Medicine, National Defense Medical University, Taipei, Taiwan, ROC. [3]Military Digital Medical Center, Tri-Service General Hospital, National Defense Medical University, Taipei, Taiwan, ROC. [4]Department of Neurological Surgery, Tri-Service General Hospital, National Defense Medical University, Taipei, Taiwan, ROC. [5]Division of Nephrology, Department of Internal Medicine, Tri-Service General Hospital, National Defense Medical University, Taipei, Taiwan, ROC.
[✉]E-mail: xup6fup@mail.ndmctsgh.edu.tw

**Glossary**

| | | | |
|---|---|---|---|
| **AF (Atrial Fibrillation)** | A common irregular heart rhythm that increases the risk of stroke and other complications. | **Hypokalemia** | A low level of potassium in the blood, also producing distinct ECG patterns. |
| **AI (Artificial Intelligence)** | Computer systems capable of learning patterns and making predictions from data, often exceeding human ability in specific tasks. | **LVD (Left Ventricular Dysfunction)** | A condition where the left ventricle does not pump blood effectively, often leading to heart failure. |
| **Alert Fatigue** | A situation where clinicians become desensitized to frequent alerts, reducing responsiveness to important warnings. | **NSTEMI (Non-ST-Elevation Myocardial Infarction)** | A heart attack without classical ST-segment elevation on ECG, often more difficult to diagnose. |
| **Arrhythmia** | An abnormal heart rhythm, which can be too fast, too slow, or irregular. | **OMI (Occlusion Myocardial Infarction)** | A newer classification that emphasizes artery blockage confirmed by angiography, rather than ECG patterns alone. |
| **Deep Learning** | A type of AI algorithm inspired by neural networks that automatically identifies complex patterns in large datasets. | **Opportunistic Screening** | The use of routine tests (like ECG) to detect unrelated hidden diseases through secondary analysis. |
| **Digital Biomarker** | A health indicator derived from digital data (e.g., ECG signals analyzed by AI). | **Phenome-wide Association Study** | An approach that links genetic or AI-derived signals with a wide range of diseases or traits. |
| **Door-to-Balloon Time** | The time between hospital arrival and opening of a blocked artery with angioplasty, a key metric in heart attack care. | **Positive Predictive Value** | The likelihood that a patient with a positive test result truly has the disease. |
| **ECG (Electrocardiogram)** | A non-invasive test recording the heart's electrical signals, widely used in cardiovascular diagnosis. | **Previvor** | An apparently healthy person identified by AI as being at high risk of developing a disease in the future. |
| **EF (Ejection Fraction)** | The percentage of blood pumped out of the left ventricle with each heartbeat, used to measure heart function. | **RCT (Randomized Controlled Trial)** | A clinical study design where participants are randomly assigned to intervention or control groups to evaluate treatment effects. |
| **Electrolyte Imbalance** | Abnormal levels of substances like potassium or calcium in the blood that can cause ECG changes. | **Sensitivity** | The ability of a test to correctly identify patients with a disease (true positive rate). |
| **Explainable AI** | AI methods designed to make predictions understandable to humans, increasing trust and transparency. | **Specificity** | The ability of a test to correctly identify patients without a disease (true negative rate). |
| **Generalizability** | The extent to which an AI model performs well across different patient groups, hospitals, or devices. | **STEMI (ST-Elevation Myocardial Infarction)** | A severe type of heart attack characterized by a specific ECG pattern indicating complete blockage of a coronary artery. |
| **Heart Failure** | A chronic condition where the heart cannot pump enough blood to meet the body's needs. | **Systematic Screening** | The use of a cheap tests (like ECG) to detect underdiagnosed diseases with a further confirmatory test for the positive finding. |
| **Hyperkalemia** | A high level of potassium in the blood, potentially life-threatening, with characteristic ECG changes. | **Wearable Device** | |

even experienced clinicians can overlook subtle yet clinically important patterns. Moreover, existing ECG data may have undiscovered knowledge of clinical importance (Yang et al, 2015). In this review, recent developments in AI-enabled ECG are summarized, existing evidence is integrated, and future research directions are proposed.

## ECG interpretation research: pre- and post-AI era

ECG, developed over a century ago, records ionic currents generated by transmembrane ion fluxes across myocardial and adjacent cells. Electrical impulses originating from the sinoatrial node propagate through the atrioventricular node and the His-Purkinje system to synchronize atrial and ventricular contractions,

ensuring effective systemic perfusion. The resulting time-dependent voltage fluctuations correspond to specific phases of the cardiac cycle, enabling electrophysiological characterization of cardiac function. The characteristics of ECG waveforms and intervals vary based on factors such as age, sex, ethnicity, and pathological cardiac conditions (Rautaharju et al, 2014).

Researchers and clinicians have identified disease-specific ECG features that enhance the diagnosis of cardiac disorders. For example, left ventricular hypertrophy (LVH) increases ion current amplitudes and alters repolarization, producing prominent R waves, ST-segment depression, and T-wave inversion in precordial leads. Diagnostic criteria for LVH, such as the Sokolow-Lyon voltage criteria (Sokolow and Lyon, 2001), Cornell voltage criterion (Devereux et al, 1984), and Romhilt-Estes point score system (Romhilt and Estes, 1968), correlate these ECG findings with imaging-confirmed left ventricular mass and are widely used in ECG analysis. ECG is crucial for detecting ischemic changes in acute myocardial

infarction (AMI), particularly ST-segment elevation myocardial infarction (STEMI). Normally in an isoelectric state, the ST segment shifts during regional ischemia, typically from acute coronary occlusion, due to altered membrane potentials and phase 0 dynamics, creating voltage gradients reflected as ST-segment deviations. ST-segment elevation combined with clinical symptoms defines the diagnostic criteria for STEMI (Thygesen et al, 2018). The pattern of ST-segment deviations according to ECG enables the ischemic territory supplied by the occluded coronary artery to be identified. Atrial fibrillation (AF), a common arrhythmia on ECGs, is triggered by ectopic atrial beats from the myocardial sleeves of the pulmonary veins. These premature impulses override sinoatrial node activity, producing disorganized atrial conduction and the absence of regular P waves. AF typically presents as paroxysmal but progresses to persistent forms as structural and histopathologic changes in the left atrium promote reentry and sustain arrhythmia (McCauley et al, 2024). Underlying conditions, including hypertension, obesity, and autonomic dysfunction, contribute to left atrial fibrosis and hypertrophy, oxidative stress, systemic inflammation, and altered atrial repolarization (Goette et al, 2016; Staerk et al, 2017), driving pathological changes in atrial electrical activity and progression from paroxysmal to persistent AF. Fundamentally, these cellular and tissue alterations in AF produce hallmark ECG features, which are critical for diagnosis and risk stratification.

In the era of AI, the paradigm of scientific research has shifted. The application of AI enables the characterization of relationships between ECG waveforms and phenotypes without relying on mechanistic studies or predefined hypotheses. Moreover, with sufficient labeled samples, the analysis can be performed directly using AI (Wang et al, 2023). The success of AlphaGo-Zero in the game of Go is one of the most well-known examples of a paradigm shift in the AI era (Silver et al, 2017). Modern AI no longer depends on human guidance or pre-existing domain knowledge; instead, it might achieve superhuman performance through self-play and the creation of extensive training datasets. A similar transformation is occurring in ECG analysis, where the application of AI can uncover patterns beyond traditional physician-driven knowledge. A notable example is AI-ECG detection of low ejection fraction (EF), an association revealed by deep learning models trained on tens of thousands of ECG data records (Attia et al, 2019b). As illustrated in Fig. 1A, ECG interpretation research is shifting from a hypothesis-driven approach to a data-driven paradigm. This shift is redefining how ECG analysis is applied in clinical practice, moving beyond established pathophysiological theories to leverage large-scale data for uncovering novel diagnostic insights, which holds the potential to significantly enhance early detection and risk stratification, ultimately improving patient outcomes. Building upon this paradigm shift, AI is not only redefining how patterns are detected but also how routine ECGs are repurposed in clinical workflows. As shown in Fig. 1B, AI can serve as a secondary analyst that complements conventional interpretation, providing diagnostic support, enhancing opportunistic screening, and prompting timely clinical interventions. This integration transforms the ECG from a static diagnostic tool into a dynamic platform for both early disease detection and proactive patient management.

## The changes of traditional ECG applications after AI-era

Various ECG diagnostic criteria exist for cardiovascular diseases, although their clinical utility depends on condition-specific accuracy. The primary ECG criterion for diagnosing AF is the presence of an irregularly irregular QRS complex without visible P waves. However, according to previous studies, less experienced physicians or general practitioners exhibit a sensitivity of 80% and a specificity of 92% when diagnosing AF based on ECG alone (Mant et al, 2007), highlighting diagnostic variability and the need for improved training or automated support. In STEMI, relying solely on ST-segment elevation for diagnosing coronary occlusion is limited by suboptimal sensitivity, reported at only 43.6%, despite a high specificity of 96.5% (de Alencar Neto et al, 2024), necessitating adjunctive markers, including reciprocal ST-segment depression, echocardiographic evidence of regional wall motion abnormalities, and elevated serum troponins. Furthermore, similar ST-segment elevation patterns may also be observed in other conditions, such as acute pericarditis, acute myocarditis, or stress cardiomyopathy (Pollak and Brady, 2012), potentially leading to misdiagnosis and unnecessary invasive procedures.

ECG also serves as an essential diagnostic support tool for identifying structural cardiac abnormalities, including left atrial enlargement (Lacalzada-Almeida et al, 2018), LVH (Levy et al, 1990), and acute pulmonary embolism (Digby et al, 2015). Additionally, extracardiac factors, such as electrolyte imbalances, particularly abnormal potassium or calcium levels, can lead to alterations in the QT interval, T-wave voltage, and axis deviation (Libby, 2021). Despite high specificity (95.1–98.8%), the low sensitivity (6.9–22.4%) of ECG criteria for LVH significantly limits their standalone diagnostic utility (Bacharova et al, 2015; Levy et al, 1990). Combining multiple LVH ECG criteria might improve sensitivity with minimal loss of specificity (Okin et al, 2017). Similarly, hyperkalemia produces a characteristic sequence of ECG changes, beginning with T-wave peaking and QT interval shortening and progressing to QRS widening and P-wave attenuation as serum potassium levels rise. However, studies have reported that the sensitivity and specificity of clinicians' use of ECG criteria for detecting hyperkalemia range from 18.0% to 52.0% and from 85.0% to 86.0%, respectively (Montague et al, 2008; Wrenn et al, 1991). Certain cardiopulmonary diseases present only with nonspecific ECG findings. Notably, aortic dissection and pneumothorax may disrupt systemic circulation and cardiac positioning, resulting in nonspecific ST-T changes or axis deviations (Arsh et al, 2024; Costin et al, 2018; Senthilkumaran et al, 2011). ECG abnormalities in these conditions, largely identified through registries and cohort studies, are nonspecific and often indistinguishable from other cardiac or noncardiac disorders. Consequently, the inherent limitations of conventional ECG criteria, which are influenced by the pathophysiology of these diseases, restrict the utility of ECG as a frontline screening tool for selected cardiovascular and pulmonary conditions.

In the AI era, traditional ECG analysis has been significantly transformed by the integration of deep learning algorithms that offer enhanced diagnostic precision. Whereas conventional criteria for conditions such as AMI, pericarditis, or electrolyte imbalances are subject to inter-observer variability, recent studies have demonstrated that AI-powered systems can detect subtle ECG features with consistently higher accuracy. These algorithms serve as decision support tools that assist physicians in earlier identification of critical conditions, potentially improving patient outcomes (Chang et al, 2021; Chen et al, 2023b; Hannun et al, 2019; Kwon et al, 2020b; Lee et al, 2022a; Lin et al, 2020; Liu et al, 2021c; Liu et al, 2022a; Liu et al, 2022b). Table 1 summarizes the reported

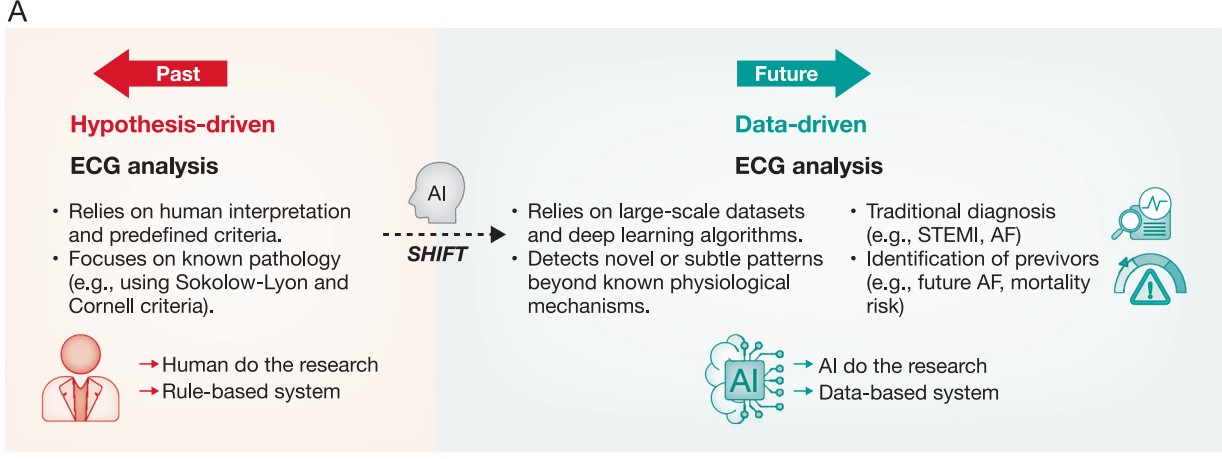

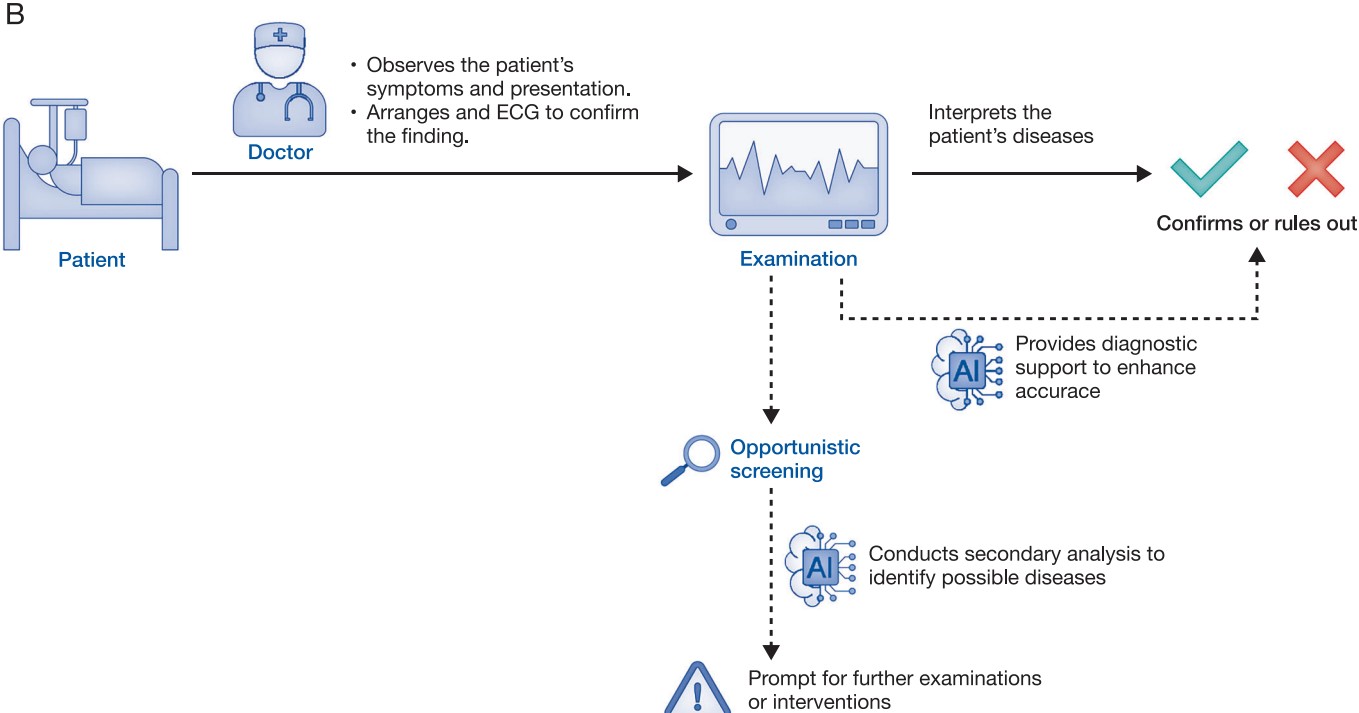

**Figure 1. Paradigm shift from conventional to AI-driven ECG analysis.**

Panel (**A**) illustrates the paradigm shift in ECG analysis caused by the use of AI. Traditionally, a hypothesis-driven approach is used to establish criteria, and human interpretations are used to identify known pathological markers. In contrast, the data-driven approach harnesses large-scale datasets and deep learning algorithms to detect subtle or novel patterns in ECG signals—those that may not be readily apparent through conventional clinical reasoning. These signatures are used to diagnose conditions such as left ventricular dysfunction and atrial fibrillation and identify "previvors," individuals with a hidden yet elevated risk for future adverse events. This transformation enables earlier detection of abnormalities, improved risk stratification, and an expanded understanding of cardiac electrical activity beyond historically defined diagnostic criteria. Panel (**B**) visually depicts the shift from conventional ECG interpretation to an integrated AI-driven approach (dashed line), emphasizing how the use of AI serves as a powerful adjunct to traditional methods (solid line) in enhancing diagnostic accuracy and patient care.

diagnostic performance from these representative studies comparing AI-based algorithms with human interpretation. Importantly, in these human–machine competitions, the AI models consistently exhibited higher sensitivity and specificity than human evaluators. These improvements highlight the potential of AI-enhanced ECG analysis to augment clinical decision making and streamline patient care by providing rapid, objective, and highly reproducible assessments of complex ECG signals.

## Feature representations in AI-ECG beyond conventional morphology

Research into algorithms for the automatic detection of ECG abnormalities has been ongoing for decades. Major breakthroughs began to emerge only after 2015, largely because traditional analytic methods were limited—they modeled the cardiac cycle merely as a concatenation of P–QRS–T segments, assuming that wave amplitudes and intervals alone

**Table 1. Comparison of the accuracy of ECG interpretation between humans and AI.**

| Diseases and study | Human expert€ | Deep learning-based AI |
|---|---|---|
| Atrial fibrillation and flutter (Hannun et al, 2019) | SENS: 71.2 ± 8.4<br>SPEC: 94.1 ± 3.3 | AUROC: 97.3<br>SENS: 86.1<br>SPEC: 94.1 |
| Trigeminy (Hannun et al, 2019) | SENS: 81.8 ± 19.7<br>SPEC: 99.5 ± 0.3 | AUROC: 99.8<br>SENS: 96.6<br>SPEC: 99.5 |
| Atrioventricular block (Hannun et al, 2019) | SENS: 72.8 ± 8.5<br>SPEC: 98.1 ± 0.8 | AUROC: 98.8<br>SENS: 85.8<br>SPEC: 98.1 |
| Left ventricular hypertrophy (Kwon et al, 2020b) | SENS: 28.7 ± 9.0<br>SPEC: 95.1 ± 1.6 | AUROC: 86.8<br>SENS: 49.6<br>SPEC: 93.6 |
| Hyperkalemia ($K^+ \geq 5.5$ mEq/L) (Lin et al, 2020) | SENS: 23.5 ± 14.4<br>SPEC: 98.2 ± 3.1 | AUROC: 95.8<br>SENS: 67.5<br>SPEC: 97.8 |
| Hypokalemia ($K^+ \leq 3.5$ mEq/L) (Lin et al, 2020) | SENS: 48.2 ± 10.0<br>SPEC: 78.1 ± 3.6 | AUROC: 92.6<br>SENS: 67.5<br>SPEC: 93.3 |
| Digoxin toxicity (Chang et al, 2021) | SENS: 84.6 ± 10.9<br>SPEC: 80.2 ± 14.1 | AUROC: 92.9<br>SENS: 84.6<br>SPEC: 96.6 |
| STEMI (Liu et al, 2021c) | SENS: 83.1 ± 8.7<br>SPEC: 86.2 ± 7.7 | AUROC: 97.6<br>SENS: 89.7<br>SPEC: 94.6 |
| Pneumothorax (Lee et al, 2022a) | SENS: 35.1 ± 29.0<br>SPEC: 67.9 ± 28.5 | AUROC: 95.7<br>SENS: 94.7<br>SPEC: 88.1 |
| Pericarditis (Liu et al, 2022b) | SENS: 49.0 ± 4.6<br>SPEC: 91.9 ± 4.6 | AUROC: 94.4<br>SENS: 76.5<br>SPEC: 100.0 |
| Pulmonary embolism (Chen et al, 2023b) | SENS: 62.5<br>SPEC: 64.5 | AUROC: 76.5<br>SENS: 70.8<br>SPEC: 69.7 |
| Aortic dissection (Liu et al, 2022a) | SENS: 45.9 ± 23.2<br>SPEC: 63.2 ± 22.8 | AUROC: 91.8<br>SENS: 81.4<br>SPEC: 87.7 |

Importantly, these accuracies are derived from specific human–machine competition datasets and may not reflect real-world performance. For example, in the STEMI human–machine competition, nearly half of the non-STEMI patients were actually NSTEMI cases, which increased the complexity of the competition (Liu et al, 2021c). Moreover, this head-to-head evaluation was conducted on case-enriched test sets. Threshold-dependent, prevalence-sensitive metrics (PPV/NPV) are intentionally not reported here; see Table 2 (diagnostic) and Table 3 (prognostic) for deployment metrics under real-world prevalence. €: The performance (%) of human experts was represented as the mean accuracy (±standard deviation) across all participating attending physicians.
*SENS* sensitivity, *SPEC* specificity, *AUROC* area under receiver operating characteristic curve.

defined cardiac function. Consequently, most early studies emphasized handcrafted feature extraction followed by classical machine-learning methods—such as support vector machines, random forests, or gradient boosting—for disease prediction, including arrhythmia classification (Zabihi et al, 2017). However, the diagnostic performance of these approaches rarely approached the accuracy of cardiologists (Guglin and Thatai, 2006). This gap reflects the fact that expert interpretation often relies on subtle morphological variations across ECG waveforms—features too numerous and nuanced to be readily captured by conventional algorithms—thereby limiting their translation into clinical practice.

The rapid advances in deep learning changed this trajectory. Landmark innovations in computer vision, including AlexNet in 2012 (Krizhevsky et al, 2012) and ResNet in 2016 (He et al, 2016), demonstrated the unprecedented capability of neural networks to automatically extract informative features from raw data. Following these developments, a surge of medical AI studies emerged after 2017, leveraging deep architectures to re-analyze existing clinical datasets (Litjens et al, 2017). By removing the dependency on predefined feature engineering, deep learning enabled models to discover discriminative ECG representations directly from data (Siontis et al, 2021). When coupled with large annotated repositories, these methods were shown to achieve—and in some tasks surpass—the diagnostic performance of human experts (LeCun et al, 2015). Consequently, recent advances in AI-ECG research have transformed the prediction and classification of diseases based on electrocardiographic data.

## Emerging applications of ECG driven by AI

The applications of ECG are rapidly expanding beyond traditional diagnostic criteria, propelled by the advent of AI. Previously,

experts recognized that left ventricular dysfunction (LVD) was associated with ECG findings reflecting its underlying causes; for example, increased QRS voltage in poorly controlled hypertension or hypertrophic cardiomyopathy, low QRS voltage in infiltrative diseases or pericardial effusion (Libby, 2021), and Q waves indicative of ischemic heart disease. Although ECG abnormalities offer insight into heart failure etiology and inform treatment strategies, they are insufficient for definitive LVD diagnosis. A landmark study demonstrated that an AI-ECG model can accurately screen for asymptomatic LVD even when such subtle abnormalities are indiscernible to human experts (Attia et al, 2019b). This breakthrough underscores the transformative potential of AI to address existing diagnostic gaps and to broaden the clinical utility of ECG. Subsequently, independent teams from various countries have developed similar AI-ECG models, achieving area under the receiver operating characteristic curves (AUROCs) exceeding 0.9 in detecting low EF (Chen et al, 2022a; Chen et al, 2022b; Cho et al, 2020; Ferreira et al, 2025; Kwon et al, 2019; Vaid et al, 2021). By leveraging deep learning algorithms, clinicians can now identify early signs of LVD, allowing timely interventions that may prevent the progression of overt heart failure. Accordingly, research is increasingly shifting its focus toward developing and refining AI applications in ECG analysis, thereby expanding the diagnostic spectrum to include conditions that were previously challenging to detect using conventional criteria.

With the breakthrough of low EF detection, the application of AI-ECG technology is expanding beyond traditional diagnostic methods. Specifically, another turning point study demonstrated that AI models could accurately identify hidden AF during sinus rhythm (Attia et al, 2019d), highlighting the diagnostic power of AI in identifying clinically important but previously underrecognized patterns. Subsequent studies have demonstrated diverse AI-ECG applications in detecting valvular heart diseases (Lin et al, 2024e), diastolic function (Lee et al, 2024), pulmonary arterial hypertension (Liu et al, 2025a), hypertrophic cardiomyopathy (Ko et al, 2020), left atrium enlargement (Lou et al, 2022), elevated brain natriuretic peptide (Liu et al, 2023), cardiac amyloidosis (Grogan et al, 2021), coronary artery disease (Awasthi et al, 2023), diabetes mellitus (Lin et al, 2021b), chronic kidney disease (Holmstrom et al, 2023), dyscalcemia (Lin et al, 2022b), anemia (Kwon et al, 2020a), hypoalbuminemia (Lee et al, 2022c), hyperthyroidism (Lin et al, 2024a), thyrotoxic periodic paralysis (Lin et al, 2021a), and cirrhosis of the liver (Ahn et al, 2022), etc. Collectively, these studies confirm the growing role of AI-ECG as a multipurpose diagnostic tool that not only revolutionizes the understanding of diseases traditionally linked to heart function but also opens new frontiers in broader medical applications, highlighting the potential for AI-ECG to enhance diagnostic accuracy and clinical decision making across a wide range of pathologies.

The advancement of AI also holds the potential to redefine existing disease classifications and drive transformative changes in clinical practice. Traditionally, AMI is classified as STEMI or non-ST-elevation myocardial infarction (NSTEMI) based on whether ECG findings meet established ST-elevation criteria (Thygesen et al, 2018). However, recent studies have raised concerns about this classification (Meyers et al, 2021). The STEMI label identifies only approximately 43.6% of patients with true total coronary occlusion, potentially missing a significant number of individuals

who would benefit from timely revascularization (de Alencar Neto et al, 2024). Therefore, a growing movement within cardiology embraces the occlusion myocardial infarction (OMI) paradigm, a transformative shift that redefines acute coronary events based not on ECG patterns alone, but on the underlying presence of coronary occlusion (McLaren et al, 2024). OMI is defined by the presence of flow-limiting coronary obstruction observed during coronary angiography rather than solely by ECG findings. There is increasing interest in the development of new diagnostic tools that can accurately detect OMI. By training models on large datasets of paired ECG and corresponding angiographic findings, AI was used to detect OMI directly from ECG signals with greater accuracy than current ST-elevation criteria (Al-Zaiti et al, 2023; Büscher et al, 2025). A head-to-head comparison between AI-based OMI detection, conventional STEMI criteria, and expert cardiologists shows that AI achieved a sensitivity of 0.806 and specificity of 0.937, substantially outperforming STEMI criteria (sensitivity 0.325, specificity 0.977) and exceeding expert cardiologists (sensitivity 0.730, specificity 0.957) (Herman et al, 2024). These findings provide direct evidence that AI-ECG can more reliably identify patients with acute coronary occlusion than current standard-of-care criteria, highlighting the potential clinical ramifications of AI-guided triage. While explainability remains an important challenge, this approach could significantly reshape the future management of AMI. In a scenario in which a standardized AI system is deployed nationally or globally, its interpretation of OMI from AI-ECGs could function as a digital biomarker, supporting cardiologists in clinical decision making. As digital biomarkers become more prevalent in diagnosis, the role of AI-ECG in clinical practice is expected to expand substantially and steadily.

Table 2 summarizes the performance metrics of several prospective studies evaluating four of the most representative emergent diagnostic applications of AI-ECG: low EF (Adedinsewo et al, 2024a; Attia et al, 2019c; Liu et al, 2024; Tsai et al, 2025; Yao et al, 2021), paroxysmal AF (Masumura et al, 2025; Noseworthy et al, 2022), hypertrophic cardiomyopathy (Desai et al, 2025; Love et al, 2025), and AMI (Lee et al, 2025; Lin et al, 2024b). Direct comparison across studies is challenging due to heterogeneity in model architecture, choice of operating thresholds (favoring either high sensitivity or high positive predictive value), variation in disease definitions, and differences in study populations. Notably, most prospective investigations to date have been conducted at the original development sites, underscoring the critical lack of external, multi-center prospective validation. Among the four applications, studies of paroxysmal atrial fibrillation consistently demonstrated low PPVs (7.3–7.6%) (Masumura et al, 2025; Noseworthy et al, 2022), likely reflecting the extremely low prevalence of the condition. Similarly, investigations of low EF conducted in outpatient settings yielded PPVs of only 4.9–6.9% (Liu et al, 2024), in stark contrast to markedly higher values (32.0–83.3%) observed in higher-prevalence cohorts (Adedinsewo et al, 2024a; Attia et al, 2019c; Tsai et al, 2025; Yao et al, 2021). These findings highlight the importance of carefully selecting the target population when implementing AI-ECG in practice. The large-scale deployment of AI-ECG is most likely to achieve clinical impact when applied in populations with an anticipated higher baseline prevalence of the target disease. Failure to account for these prevalence-dependent variations risks overstating the

**Table 2. Representative AI-driven emergent diagnostic ECG applications in prospective studies.**

| Diseases and study | Population | AUROC | SENS | SPEC | PPV | NPV |
|---|---|---|---|---|---|---|
| EF ≤ 35% (Attia et al, 2019c) | 6008 patients who had an echocardiogram within a year from the ECG | 0.918 (internal) | 82.5 (internal) | 86.8 (internal) | 32.0 (internal) | 98.5 (internal) |
| EF ≤ 50% (Yao et al, 2021) | 343 patients with high-risk of LVD flagged by AI-ECG and follow-up echocardiogram in the intervention group | | | | 71.1 (internal) | |
| EF < 45% (Adedinsewo et al, 2024a) | 100 pregnant and postpartum women (up to 12 months following delivery) | 1.000 (internal) | 100.0 (internal) | 98.9 (internal) | 83.3 (internal) | 100.0 (internal) |
| EF ≤ 35% (Liu et al, 2024) | 7645 patients without history of heart failure and prior echocardiogram in outpatient department had an echocardiogram within 28 days from the ECG | 0.984 (internal) 0.945 (external) | 92.6 (internal) 63.6 (external) | 93.8 (internal) 93.7 (external) | 6.9 (internal) 4.9 (external) | 100.0 (internal) 99.8 (external) |
| EF ≤ 50% (Tsai et al, 2025) | 1129 hospitalized patients who had an echocardiogram within 30 days from the ECG in the intervention group | | 76.0 (internal) | 85.2 (internal) | 34.2 (internal) | 97.2 (internal) |
| Paroxysmal AF (Noseworthy et al, 2022) | 1003 patients had 12-lead ECG and continuous ambulatory heart rhythm monitor | | 88.9 (internal) | 38.4 (internal) | 7.6 (internal) | 98.4 (internal) |
| Paroxysmal AF (Masumura et al, 2025) | 3362 adults had a normal 12-lead ECG without AF history | 0.750 (external) | 81.8 (external) | 67.5 (external) | 7.3 (external) | 99.2 (external) |
| HCM (Desai et al, 2025) | 45,873 patients with compliant ECG | | 80.1 (external) | 99.1 (external) | 54.9 (external) | 99.7 (external) |
| HCM (Love et al, 2025) | 217 patients with high-risk of HCM flagged by AI-ECG and follow-up examinations | | | | 7.8 (external) | |
| STEMI with total occlusion (Lin et al, 2024b) | 21,612 patients presented at emergency department and inpatient department in the intervention group | | 87.0 (internal) | 99.8 (internal) | 62.0 (internal) | 100.0 (internal) |
| AMI (Lee et al, 2025) | 8493 patients presented at 18 emergency departments with suspected AMI | 0.878 (external) | 76.7 (external) | 84.8 (external) | 53.6 (external) | 94.1 (external) |

Because accuracy varies with the number of leads, only the results of studies using 12-lead ECG are included here. The word "internal" or "external" in brackets indicates whether the field of prospective application is consistent with the source of model development. Studies using only ECG definition of STEMI will be excluded.
AUROC area under receiver operating characteristic curve, SENS sensitivity, SPEC specificity, PPV positive predictive value, NPV negative predictive value, EF ejection fraction, AF atrial fibrillation, HCM hypertrophic cardiomyopathy, STEMI ST-segment elevation myocardial infarction, AMI acute myocardial infarction.

generalizability of AI-ECG and may ultimately undermine its translation into routine clinical care.

AI-ECG has emerged not only as a diagnostic tool but also as a powerful method for risk stratification and is capable of identifying "previvors", defined as individuals who appear healthy but harbor a high predisposition to disease (Attia et al, 2021). For example, as demonstrated in the first breakthrough LVD study (Attia et al, 2019b), AI-ECG was used to predict the risk of future LVD in individuals with normal EF. Intriguingly, those with false-positive findings were found to have a five-fold increased risk of developing LVD over the subsequent five years. Similar findings have been reported in other AI-ECG diagnostic models, where false-positive predictions of dyskalemia were linked to abnormal ECG rhythms and an increased risk of mortality and hospitalization (Lin et al, 2022a). Moreover, false-positive predictions across various diagnostic models have consistently been linked to poorer prognoses (Chang et al, 2022; Chen et al, 2022b; Lee et al, 2022b; Lee et al, 2022c; Lin et al, 2022b; Lin et al, 2024a; Lin et al, 2021b; Lin et al, 2024e; Lou et al, 2022). This observation introduces the concept of "disease previvors" and suggests that AI-ECG models, by detecting subtle electrophysiological signatures, may be used to predict future adverse outcomes. Notably, direct prediction models, such as those that forecast future heart failure, have shown even greater accuracy than diagnostic models (Dhingra et al, 2025). In addition to heart failure, AI-ECG has been leveraged to predict mortality (Raghunath et al, 2020), major cardiovascular adverse events (Lin et al, 2025), AF (Khurshid et al, 2022), hypertension (Sau et al, 2025a), and the potential need for pacemaker implantation (Hung et al, 2024). Many of these prediction models appear to share common features, enabling a model designed to forecast mortality to also predict a broad range of future clinical events   (Sau et al, 2024a; Tsai et al, 2023). Collectively, these studies reinforce the transformative potential of AI-ECG in redefining risk stratification and preventive cardiology, illustrating the potential of a paradigm shift that re-evaluates the role of routine ECG screening in both clinical and community settings.

Although AI-ECG has been shown to associate with long-term outcomes, its performance in prognostic tasks often deteriorates substantially during external validation, in contrast to diagnostic applications that have withstood external and even prospective evaluation. Table 3 summarizes two of the most representative prognostic tasks: all-cause mortality (Sau et al, 2024a; Tsai et al, 2023) and atrial fibrillation (Khurshid et al, 2022; Sau et al, 2025b). Across these published studies, external validation was associated with a decline in performance of 5–20% in AUROC. Moreover, threshold cutoffs derived from internal cohorts frequently failed to generalize, leading to substantial shifts in sensitivity, specificity, positive predictive value, and negative predictive value during external testing. For example, a model originally developed at an academic medical center achieved a specificity of 75.8%, yet dropped to only 32.8% when applied to the UK Biobank (Sau et al, 2024a). Unlike positive predictive value, specificity is not directly influenced by disease prevalence; thus, such variability suggests that population-specific thresholds may be required—a practical challenge for large-scale implementation. Nevertheless, despite these performance losses, most external validations still achieved AUROC values exceeding 0.65, supporting the notion that ECGs do indeed carry prognostic information for long-term outcomes. These findings highlight both the promise and the limitation of

AI-ECG: while a standard 10-second tracing may harbor sufficient signals to predict events more than a year into the future, further research is needed to elucidate the sources of performance degradation across populations and to develop strategies that ensure robust generalizability in prognostic applications. Unless these limitations are addressed, the role of AI-ECG in long-term risk prediction is likely to remain confined to research settings, falling short of broad clinical translation.

AI-ECG technology has shown remarkable potential in detecting various health conditions, extending beyond traditional cardiovascular disease diagnosis and prediction. AI models have been trained to identify key demographic factors such as sex (Attia et al, 2019a) and age (Chang et al, 2022), patterns that cannot be explained by human experts. Furthermore, AI-ECG has demonstrated the ability to identify biometric characteristics (Mangold et al, 2024). These capabilities suggest that AI-ECG may reveal insights beyond diagnosis, offering a deeper view of an individual's overall physiological state. The expanding applications of AI-ECG call for a reimagining of clinical diagnostics, shifting toward data-driven discovery through unbiased hypothesis-free exploration (Fig. 2). A compelling example of this shift was reported in a recent study (Lou et al, 2023), where investigators leveraged EHR systems to curate clinical datasets, enabling comprehensive evaluation of AI-ECG, which achieved an AUROC > 0.7 across more than 70 clinical outcomes. This data-driven approach can reveal unexpected links between ECG patterns and a wide range of diseases, extending far beyond traditional ECG diagnostics. Extending this paradigm, a recent phenome-wide association study of AI-ECG has evaluated over 600 diseases to enable the prediction of a broad spectrum of future clinical conditions, highlighting the expanding potential of AI-ECG for predicting numerous clinical outcomes, particularly in the circulatory, respiratory, and endocrine/metabolic systems (Friedman et al, 2025). Thus, the value of unbiased open-ended exploration is emphasized to fully harness the potential of AI-ECG, promoting novel uses of ECG data for diagnosis and prognosis.

# AI-ECG in diagnostic support and opportunistic screening

While high diagnostic accuracy is essential, it does not inherently lead to better clinical outcomes or healthcare value. A key challenge in adopting AI technologies is demonstrating real-world utility beyond accuracy metrics (Han et al, 2024). This is where prospective clinical trials, particular for randomized controlled trials (RCTs)—the gold standard for clinical evidence, play a pivotal role. Prospective clinical trials help investigators move beyond retrospective measurements of sensitivity and specificity, providing a structured framework to assess whether integrating AI-ECG into routine practice improves patient outcomes, reduces healthcare costs, or meaningfully changes clinical decision making. By comparing AI-guided care with standard care or alternative interventions, prospective clinical trials offer a more robust assessment of real-world performance and help uncover potential challenges in workflow integration, clinician acceptance, and patient adherence.

Here, we conducted a systematic review and identified 11 prospective clinical trials, which are summarized in Table 4. The detailed search strategy, screening process, and inclusion criteria

**Table 3.** Performance of representative AI-driven emergent prognostic ECG applications in internal and external validation studies.

| Diseases and study | Time interval | Validation type | Population | AUROC | SENS | SPEC | PPV | NPV |
|---|---|---|---|---|---|---|---|---|
| All-cause mortality (Tsai et al, 2023) | 1 year | Internal | 27,808 patients from whole hospital | 0.894 | 37.0 | 97.4 | 27.4 | 98.3 |
| All-cause mortality (Tsai et al, 2023) | 1 year | External | 33,047 patients from whole hospital | 0.858 | 39.4 | 95.7 | 31.0 | 97.0 |
| All-cause mortality (Tsai et al, 2023) | 2 years | External | 1959 patients in SaMi-Trop database | 0.695 | 43.6 | 82.6 | 18.0 | 94.4 |
| All-cause mortality (Tsai et al, 2023) | 5 years | External | 345,799 ECGs from 233,700 patients in CODE-15% database | 0.688 | 31.1 | 86.3 | 21.8 | 91.1 |
| All-cause mortality (Sau et al, 2024a) | 5 years | Internal | 434,262 patients from whole hospital | 0.831 | 75.3 | 75.8 | 62.5 | 85.2 |
| All-cause mortality (Sau et al, 2024a) | 10 years | Internal | 434,262 patients from whole hospital | 0.838 | 77.4 | 75.0 | 84.6 | 65.1 |
| All-cause mortality (Sau et al, 2024a) | 2 years | External | 1959 patients in SaMi-Trop database | 0.777 | 84.2 | 57.5 | 14.8 | 97.7 |
| All-cause mortality (Sau et al, 2024a) | 5 years | External | 345,799 ECGs from 233,700 patients in CODE-15% database | 0.803 | 72.8 | 74.0 | 31.2 | 94.4 |
| All-cause mortality (Sau et al, 2024a) | 5 years | External | 40,265 patients in UK Biobank | 0.654 | 82.0 | 32.8 | 5.2 | 97.6 |
| All-cause mortality (Sau et al, 2024a) | 5 years | External | 13,739 patients in ELSA-Brasil database | 0.731 | 70.0 | 62.7 | 3.1 | 99.2 |
| All-cause mortality (Sau et al, 2024a) | 10 years | External | 13,739 patients in ELSA-Brasil database | 0.715 | 65.7 | 63.4 | 25.1 | 90.8 |
| AF (Khurshid et al, 2022) | 5 years | Internal | 4166 patients without AF | 0.823 | | | | |
| AF (Khurshid et al, 2022) | 5 years | External | 37,963 patients without AF | 0.747 | | | | |
| AF (Khurshid et al, 2022) | 2 years | External | 41,033 patients without AF in UK Biobank | 0.705 | | | | |
| AF (Sau et al, 2025b) | 5 years | Internal | 55,619 patients from whole hospital | 0.756 | 71.7 | 66.6 | 9.9 | 97.9 |
| AF (Sau et al, 2025b) | 5 years | External | 38,892 patients without AF in UK Biobank | 0.701 | 78.7 | 46.9 | 2.3 | 99.3 |

AUROC area under receiver operating characteristic curve, SENS sensitivity, SPEC specificity, PPV positive predictive value, NPV negative predictive value, AF atrial fibrillation, LVD left ventricular dysfunction.

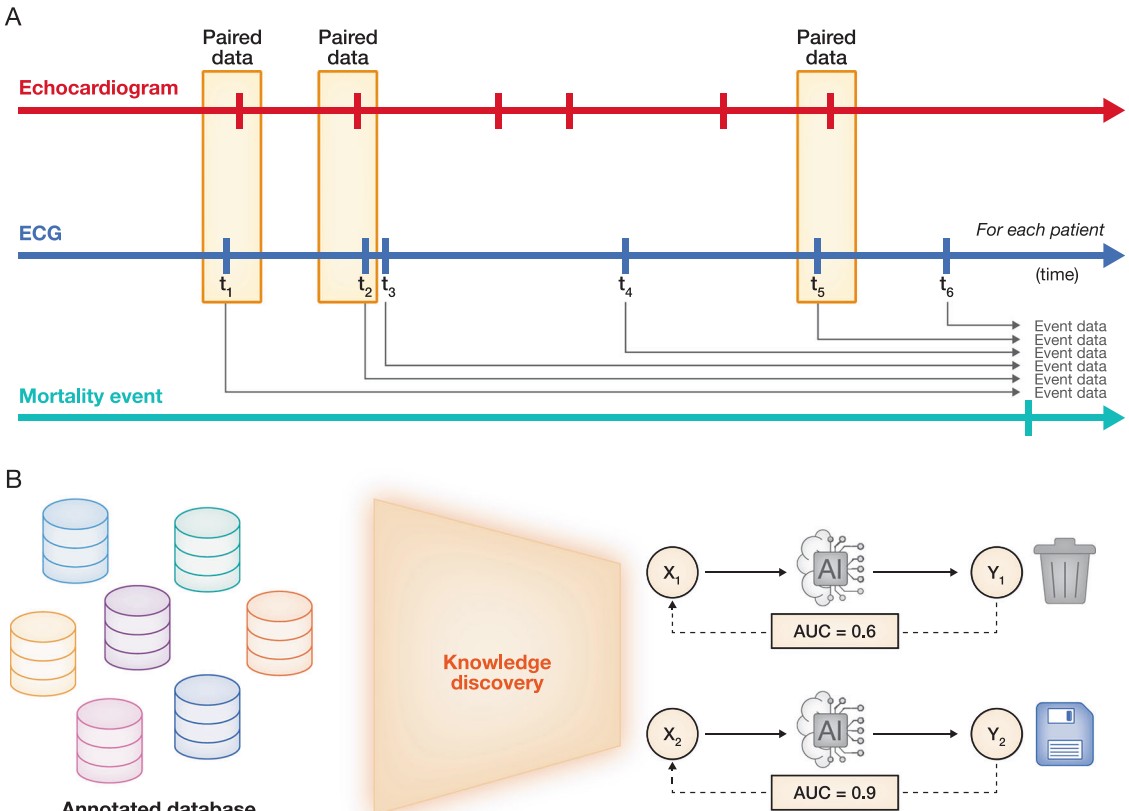

**Figure 2. Unbiased data-driven knowledge discovery in AI-ECG.**

(A) The electronic health record (EHR) contains complete information about a patient's entire life journey. We can establish paired data between ECG and specific test results using a designated time window (as illustrated in the top panel) or annotate ECG with a specific event (as shown in the bottom panel). This approach can generate thousands of annotated databases. (B) These annotated databases can then be explored for correlations using AI algorithms. If the analysis demonstrates high accuracy, the findings can be retained. If the accuracy is low, they can be discarded. This approach not only broadens the diagnostic spectrum of ECG but also lays the groundwork for precision medicine by revealing the interconnected nature of systemic pathologies.

are described in the Appendix. In ECG-based STEMI detection, the ARISE trial exemplifies the successful integration of AI-ECG into clinical practice (Lin et al, 2024b). This pragmatic RCT demonstrated that AI-ECG-assisted triage significantly reduced door-to-balloon time for STEMI patients presenting in the emergency department, with a median reduction of 14 min compared with standard care (96 min to 82 min). This reduction is significant for managing AMI, as a shorter door-to-balloon time is closely associated with lower mortality rates (Terkelsen et al, 2010). Importantly, the ARISE trial involved a comprehensive clinical pathway integrating AI-ECG alerts, wherein on-duty cardiologists received AI-ECG notifications and administered catheterization procedures. The effectiveness of this pathway likely stems from multiple factors, including a reasonable false-positive rate. It has been reported that 49–96% of clinical alerts were overridden (van der Sijs et al, 2006). The AI-ECG achieved a positive predictive value of 89.5%, markedly surpassing the Philips automated system's value of 12.9%, which likely encouraged on-duty cardiologists to respond to alerts even during late-night hours. In addition to the ARISE trial, two prospective nonrandomized studies have demonstrated the effectiveness of AI-ECG in reducing door-to-balloon time (Liu et al, 2021b; Wang et al, 2022). Overall, these studies

highlight the potential of next-generation, deep learning-based AI-ECG systems to provide diagnostic support and improve clinical outcomes.

Another promising application of AI-ECG is opportunistic screening. This approach is inspired by practices in radiology, where incidental findings on unrelated imaging often improve prognoses through early intervention (Berland et al, 2010). A classic example is incidental adrenal masses. While frontline clinicians seldom order imaging to screen for them, radiologists routinely scrutinize scans for all potential abnormalities (Mayo-Smith et al, 2017). The reanalysis of imaging data for secondary findings has gained traction in radiology, with recent RCTs showing that the use of AI can replicate radiologists' secondary analyses of chest X-rays to detect additional osteoporosis cases (Lin et al, 2024c). By adopting a similar strategy, AI-ECG systems could provide opportunistic screening for hidden or asymptomatic cardiac disorders during routine ECG examinations. By detecting subtle abnormalities beyond conventional analysis, AI can enhance early detection, enable timely intervention, and improve patient outcomes.

Building on this opportunistic screening paradigm, the EAGLE trial represents a pivotal advancement in harnessing AI for cardiac

**Table 4. Prospective studies evaluating AI-ECG on clinical impacts.**

| Diseases and study | Study design | Primary endpoint | Main study population | Key findings |
|---|---|---|---|---|
| Low EF (Yao et al, 2021) | RCT | New-onset low EF (≤50%) events within 90 days | Patients receiving routine ECGs (n. intervention = 11,573; n. control = 11,068) | New diagnosis of low EF increased from 1.6% to 2.1% (OR: 1.32, 95% CI 1.01–1.61) |
| STEMI (Liu et al, 2021a) | Non-RCT | D2B time | ED STEMI cases (n. intervention = 32; n. control = 57) | D2B time reduced from 69 to 61 min ($p = 0.037$) |
| Paroxysmal AF (Noseworthy et al, 2022) | Non-RCT | New AF diagnosis | 1003 patients with continuous monitoring (intervention group); 1003 matched patients (control group) | New diagnosis of AF significant increased from 3.6% to 10.6% in AI-ECG identified high-risk subgroup (HR: 2.85, 95% CI 1.83–4.42). In low-risk subgroup, insignificant benefit was shown (1.1% to 2.6%, $p = 0.12$) |
| STEMI (Wang et al, 2022) | Non-RCT | D2B time | ED STEMI cases (n. intervention = 68; n. control = 86) | D2B time reduced from 66 to 53 min ($p = 0.007$) |
| All-cause mortality (Lin et al, 2024d) | RCT | All-cause mortality event within 90 days | ED or IPD patients with ECGs (n. intervention = 8001; n. control = 7964) | Mortality events reduced from 4.3% to 3.6% (HR: 0.83, 95% CI 0.70–0.99) |
| STEMI with total occlusion (Lin et al, 2024b) | RCT | D2B time | ED STEMI cases (n. intervention = 67; n. control = 65) | D2B time reduced from 96 to 82 min ($p = 0.002$) |
| Low EF (Adedinsewo et al, 2024b) | RCT | Low EF (<50%) diagnosis | Patients in perinatal period (n. intervention = 587; n. control = 608) | New diagnosis of low EF increased from 2.0% to 4.1% (OR: 2.12, 95% CI 1.05–4.27) |
| Low EF (Thao et al, 2024) | Post-hoc analysis | ICER per QALY | Patients receiving routine ECGs (n. intervention = 11,573; n. control = 11,068) | AI-ECG cost-effective with ICER $27,858/QALY |
| Low EF (Tsai et al, 2025) | RCT | New-onset low EF (<50%) events within 30 days | ED or IPD patients with ECGs (n. intervention = 6840; n. control = 6791) | New diagnosis of low EF increased from 1.1% to 1.5% (HR: 1.50, 95% CI 1.11–2.03) |
| All-cause mortality (Hsieh et al, 2025) | Post-hoc analysis | Incremental cost per death averted | ED or IPD patients with ECGs (n. intervention = 8001; n. control = 7964) | ICER $59,500 per death averted (95% CI −$4,657 to $385,950) |
| Atrial fibrillation (Liu et al, 2025b) | RCT | NOAC prescription; new AF diagnosis; echocardiography; cardiology referral (within 90 days) | ED or IPD patients with AI-ECG positive cared by non-cardiologists (n. intervention = 275; n. control = 245) | More NOAC prescription from 12.0% to 23.3% (HR: 1.85, 95% CI 1.11–3.07) and new AF diagnosis from 36.0% to 47.8% (HR: 1.40, 95% CI 1.03–1.90). |

*EF* ejection fraction, *RCT* randomized controlled trial, *STEMI* ST-segment elevation myocardial infarction, *D2B* door-to-balloon, *ED* emergency department, *AF* atrial fibrillation, *NOACs* new prescription of non-vitamin K oral anticoagulants, *IPD* inpatient department, *ICER* Incremental cost-effectiveness ratio, *QALY* quality-adjusted life year.

care (Yao et al, 2021). In this cornerstone investigation, researchers integrated a deep learning model into routine ECG analysis to identify subtle electrocardiographic patterns indicative of low EF. By prospectively embedding the AI tool into clinical workflows, this study not only demonstrated robust sensitivity and specificity in detecting asymptomatic patients with subclinical LVD but also revealed how incidental findings can drive early therapeutic interventions. Importantly, this trial is the first to show that AI-ECG provides clinical benefits beyond traditional ECG applications, even without fully understood mechanisms. In addition, a post-hoc economic evaluation revealed that this strategy was cost-effective, with an incremental cost-effectiveness ratio of $27,858 per quality-adjusted life year gained (Thao et al, 2024). A subsequent RCT further demonstrated that AI-ECG–based opportunistic screening for LVD is clinically valuable in both hospitalized and emergency department populations (Tsai et al, 2025). Despite our limited understanding of how an AI model identifies signs of low EF from ECG data, the ability of AI models to detect such hidden patterns suggests a paradigm shift in clinical practice. This work highlights the transformative potential of AI-ECG systems to repurpose standard diagnostic tests into powerful, low-cost screening instruments that can seamlessly augment traditional care and ultimately improve cardiovascular outcomes.

Building on earlier advances in AI-enhanced ECG screening, a breakthrough randomized trial has demonstrated the real-world benefits of integrating AI alerts into clinical workflows (Lin et al, 2024d). In this trial, nearly 16,000 hospitalized patients underwent routine ECG assessments, with the AI system generating real-time alerts for those developing high-risk patterns. Compared with the control group, the intervention group, which received tailored clinical recommendations based on these alerts, experienced a significant reduction in 90-day all-cause mortality, most notably among patients with markedly abnormal ECG findings. An economic evaluation further confirmed that the AI-ECG alert system was cost-effective, with an incremental cost-effectiveness ratio of approximately $59,500 per death averted, strengthening its health system value (Hsieh et al, 2025). This study highlights the effectiveness of AI-driven alerts in prompting timely interventions and validates the repurposing of routine diagnostic tools as dynamic screening tools. Together with the foundational insights from the earlier EAGLE study, these findings herald a new era in personalized cardiovascular care, where the use of AI augments clinician decision making to improve patient outcomes.

In brief, the evolving AI-ECG landscape has potential not only for high diagnostic accuracy but also for transforming routine ECG into a dynamic second-opinion tool that complements conventional diagnostic methods. The ARISE trial's reduction in door-to-balloon time, the EAGLE trial's detection of subclinical LVD, and recent randomized evidence of reduced 90-day mortality highlight the real-world impact of integrating AI into clinical workflows. Moreover, a prospective, nonrandomized interventional trial demonstrated that the AI-guided approach significantly improved silent AF detection compared with standard care (Noseworthy et al, 2022). Another RCT demonstrated that AI-ECG significantly increased the diagnosis of AF in both inpatient and emergency department populations and was associated with higher long-term prescription rates of non-vitamin K oral anticoagulants (Liu et al, 2025b). These studies illustrate that AI-ECG can enhance patient triage, expedite critical interventions, and reveal incidental findings.

Notably, there are many AI-ECG applications, which still require validation through additional RCTs or prospective studies. All these findings reinforce the need for continued research to define its clinical value, optimize integration, and establish AI-based screening as a reliable element of personalized cardiovascular care in the near future.

## AI-ECG in systematic screening

AI-ECG in systematic screening represents a transformative opportunity to re-examine the role of ECG screening in asymptomatic populations. Previous studies have cast doubt on the economic viability of routine ECG screening in these groups. For example, the US Preventive Services Task Force (Curry et al, 2018) and a subsequent survey on cascades of care following incidental findings (Ganguli et al, 2019) indicated that traditional ECG screening often triggers costly follow-up tests and interventions without demonstrable clinical benefit. These findings are largely attributed to the limited scope of conventional ECG analysis, which, before the AI era, focused primarily on detecting conditions such as AF. Moreover, earlier systematic reviews have shown that community screening for AF alone does not yield favorable economic outcomes (Moran et al, 2016). Notably, the advent of AI-ECG technologies has expanded the diagnostic reach far beyond AF by revealing subtle and previously undetectable cardiac abnormalities. This enhanced capability suggests that AI-ECG could transform community screening into a more economically attractive proposition by enabling earlier interventions and more targeted care.

Recent cost-effectiveness analyses in the USA (Tseng et al, 2021) and Taiwan (Liu et al, 2024) suggest that ECG screening for low EF in asymptomatic populations may yield significant economic benefits. The differing economic outcomes between screening for low EF and AF may stem from the higher incidence of adverse events associated with low EF, leading to increased healthcare costs. Studies conducted in various countries have demonstrated that AI-ECG screening for low EF is cost effective, highlighting its broad applicability. In addition to low EF detection, AI-ECG can be used to screen for other clinically significant abnormalities, such as subtle arrhythmias and early signs of structural heart disease. This multifaceted capability enhances the overall cost effectiveness of community ECG screening, providing a robust platform for preventive cardiology. Consequently, these findings highlight the need to re-evaluate the role of ECG screening in community settings.

Interestingly, a recent study evaluated an AI-ECG screening approach for cardiomyopathies in a highly specific, high-risk population outside traditional hospital settings (Adedinsewo et al, 2024b). Although this study focused on a narrow population of pregnant and postpartum women, it provides some of the first robust evidence supporting AI-ECG screening in the community. The results showed that AI-ECG was used to detect subclinical cardiac dysfunction with high sensitivity and specificity, enabling earlier intervention in settings where conventional screening had limited reach. All the findings support that integrating AI with routine ECG assessments can transform community screening practices, not only by identifying low EF but also by uncovering other significant cardiac abnormalities. While the findings are

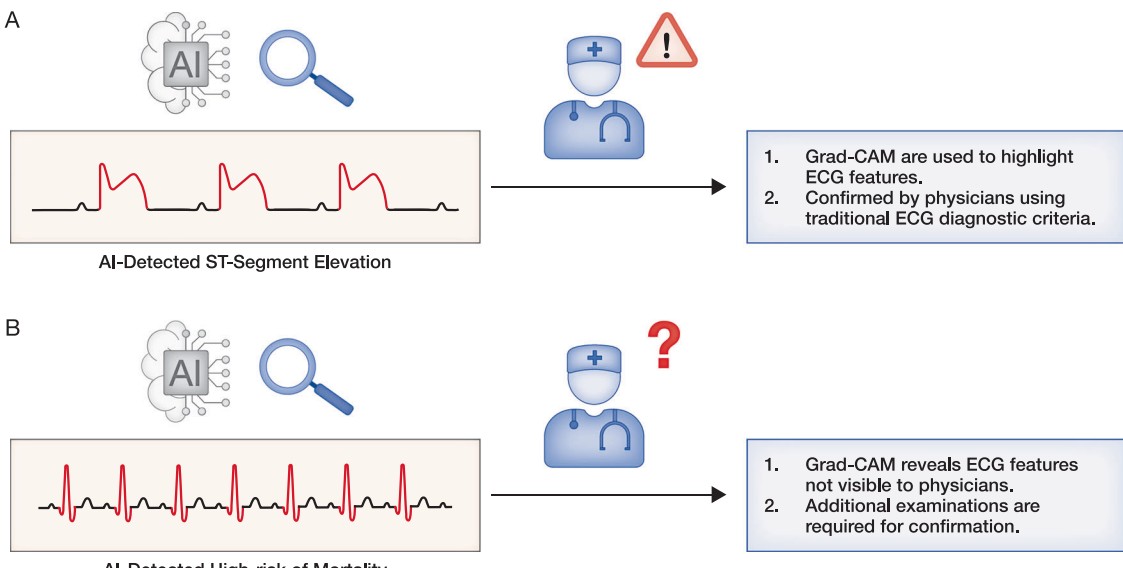

**Figure 3. Schematic illustration of Gradient-weighted Class Activation Mapping (Grad-CAM) for AI-ECG interpretability in clinical decision support.**

(A) Interpretability of an AI-ECG model in STEMI. The AI model detects STEMI, and Grad-CAM localizes the ST-segment elevation driving the prediction. Physicians can rapidly verify the finding using conventional ECG diagnostic criteria and initiate appropriate management. (B) Interpretability of an AI-ECG model for mortality risk prediction. The model identifies a patient as high-risk for mortality, with Grad-CAM indicating the QRS complex as the dominant contributing feature. However, these model-highlighted features are not readily interpretable by clinicians, limiting the ability to validate the prediction despite the application of explainability techniques.

promising, broader studies are needed to validate and generalize these results across diverse populations. Collectively, the evidence suggests that the use of AI-ECG holds substantial potential for redefining preventive cardiology in community settings, warranting a re-evaluation of ECG screening strategies in the era of AI.

## Frameworks for AI-ECG deployment

Although numerous clinical trials have demonstrated the potential of AI-ECG in real-world practice, multi-center evidence remains limited. Successful clinical adoption of AI-ECG will require a standardized operational framework to ensure consistent performance across settings. Traditionally, 12-lead ECGs are recorded over 10 s at a sampling frequency of 500 Hz and stored in standardized digital formats. ECGs acquired outside these formats may require additional preprocessing steps, introducing uncertainty and variability in model performance. Moreover, filters applied during acquisition—such as baseline wander suppression, artifact reduction, or frequency cutoffs—can significantly influence model generalizability (Armoundas et al, 2024). Similarly, some AI-ECG models are trained on ECG images, either exported directly from devices or captured externally, but these also require standardized image preprocessing to ensure uniform resolution and consistent feature representation (Sau et al, 2024b; Vaid et al, 2023). Harmonization of acquisition and preprocessing protocols is therefore critical to the reliable and scalable deployment of AI-ECG systems in clinical care.

Threshold selection represents another critical consideration when deploying AI-ECG models for decision support. Although optimal thresholds can be determined during model training, they may need to be adapted to specific clinical contexts to enhance usability and minimize alert fatigue. A commonly used approach is to define the cutoff at the Youden index on the AUROC curve, thereby maximizing the combined sensitivity and specificity. However, disease prevalence strongly shapes the positive predictive value (PPV) for any given threshold. For example, in a population with 1% prevalence of dilated cardiomyopathy, an AI-ECG model with 98.8% sensitivity and 44.8% specificity achieved a PPV of only 1.8%, whereas in a population with 5% prevalence, the PPV increased to 8.6% despite identical sensitivity and specificity (Shrivastava et al, 2021). Thus, adjusting thresholds in light of disease prevalence and incorporating precision–recall analyses may better optimize the trade-off between sensitivity and PPV, particularly for conditions with low prevalence.

While AI models can deliver accurate predictions when trained on large datasets, final clinical decisions remain the responsibility of physicians, who must integrate AI outputs with the broader patient context. To enhance interpretability, visualization techniques such as saliency maps and Gradient-weighted Class Activation Mapping (Grad-CAM) are often used to highlight ECG features emphasized by the model, thereby supporting physician judgment (Jahmunah et al, 2022; Siontis et al, 2023). However, these methods are most effective when the diagnostic criteria are visually apparent on the ECG—for example, T-wave abnormalities in dyskalemia or ST-segment deviations in STEMI (Fig. 3A, AI-ECG with STEMI). In conditions such as LVD or elevated mortality risk, the features recognized by AI-ECG models are often subtle or not readily discernible to clinicians, limiting their ability to independently verify the predictions (Fig. 3B, AI-ECG with high mortality risk). Although the utility of interpretability techniques varies across models, advancing more robust and clinically meaningful approaches to explainability is essential to foster physician confidence and ensure reliable integration of AI-ECG into clinical workflows.

Notably, individualized and protocolized care pathways, integrated with alerts generated by AI-ECG models, are essential to ensure consistent clinical impact and reproducibility across diverse healthcare settings. Determining appropriate confirmatory testing following diagnostic model predictions is critical. For example, echocardiography may be indicated for suspected LVD (Tsai et al, 2025; Yao et al, 2021), prolonged ECG patch monitoring for paroxysmal AF (Noseworthy et al, 2022), and stress testing or coronary computed tomography angiography for coronary artery disease (Lee et al, 2023). However, local healthcare resources inevitably influence the choice of confirmatory tests, and these management strategies should be validated in prospective studies to ensure reliability. For AI models that stratify future risk of adverse events, such as heart failure or mortality, early risk factor modification, closer monitoring intervals, or pre-emptive therapy may be considered, though evidence to support standardized management remains limited and patient-centered care with tailored follow-up is still required (Lin et al, 2024d). In models designed for time-sensitive conditions, such as STEMI (Lin et al, 2024b), protocols to expedite referral pathways and streamline team-based management are indispensable to maximize effectiveness. Ultimately, the integration of AI-ECG into clinical workflows will depend not only on technical accuracy but also on the establishment of evidence-based management pathways that can translate predictions into improved patient outcomes.

## Future directions and recommendations

Advances in AI have opened new avenues in ECG analysis, and future directions should address both data modality and integration challenges. Regarding the data modality, although most prior studies have focused on analyzing raw ECG signals, several institutions store ECG data as images, which makes image-based deep learning models a valuable alternative for processing these archived data. Several studies have yielded promising results in this area, although further large-scale validation is needed (Sau et al, 2024b; Vaid et al, 2023). Moreover, as cardiovascular diagnoses often require information beyond the ECG alone, the integration of additional clinical data, such as chest X-rays or biomarkers such as D-dimer, which improve the detection of conditions such as aortic dissection (Liu et al, 2022a), can further enhance diagnostic accuracy and model generalizability. In addition, the growing popularity of wearable devices has spurred research into single-lead ECG analysis. Notably, studies using Apple Watch data have demonstrated that lead I alone can be used to detect low EF (Attia et al, 2022) and hyperkalemia accurately (Chiu et al, 2024). Future research should extend these findings to other consumer devices, adapting signal processing techniques to ensure robust performance outside clinical settings.

Various physiological and technical factors, such as environmental changes, emotional fluctuations, and acute illness status, might affect ECG findings and interpretation. Acute alcohol intake has been shown to significantly decrease short-term heart rate variability in healthy subjects (Ryan and Howes, 2002). Moreover, time of day may influence ECG characteristics by modulating autonomic tone (Korpelainen et al, 1997). Therefore, establishing rigorous standards for ECG measurement, akin to the strict protocols for blood pressure assessment recommended by most cardiovascular societies, is essential to minimize variability and error (Muntner et al, 2019). In practice, standardized AI-ECG protocols remain underexplored, as most studies rely on hospital-based 12-lead ECG data with limited serial measurements in relatively stable clinical settings. To address this gap, future efforts should incorporate wearable devices to collect home ECG signals and link them with patients' concurrent physiological states. This approach would enable the development of a standardized pre-ECG protocol to ensure that ECG acquisition and interpretation, thereby enhancing the reliability and clinical utility of ECG-derived metrics in both research and routine care.

Another key future direction is exploring how to intervene in high-risk patients identified by predictive models to reduce the likelihood of adverse events. Although early studies have shown that AI-ECG-triggered emergency interventions can reduce in-hospital mortality (Lin et al, 2024d), long-term studies are needed to clarify how AI-derived risk scores can inform clinical management. For example, patients flagged by AI models as having subtle signs of LVD despite normal EF might benefit from being classified as stage A or pre-A heart failure, prompting earlier intervention. Moreover, while various AI-ECG models demonstrate excellent diagnostic performance, their "black-box" nature limits our understanding of the underlying physiological correlates. Enhancing model interpretability through explainable AI approaches could provide insights into disease mechanisms and stimulate basic research, ultimately leading to new therapeutic targets.

Additionally, the current proliferation of AI models, where a single ECG record can generate predictions from dozens of algorithms, risks overwhelming clinicians with information. Therefore, developing integrated platforms that consolidate outputs into a streamlined, actionable summary is essential. Notably, if a mortality risk model simultaneously predicts several outcomes (Sau et al, 2024a; Tsai et al, 2023), an ensemble or selection strategy should be employed to present only the most relevant information for clinical decision making. Such integration would streamline interpretation, reduce cognitive burden, and enhance the clinical utility and cost-effectiveness of AI-enhanced ECG analysis. However, incorporating these models into clinical guidelines remains challenging and requires further RCTs to validate their impact on outcomes.

Finally, many AI-ECG models are developed using training data predominantly from a single ethnic group or geographic region; consequently, their performance can degrade or become biased when applied to different populations (Chen et al, 2023a). This reflects issues of dataset shift (distributional differences between training and target populations) and model fairness, as an algorithm's accuracy may vary across demographic subgroups. In one study, for example, an ECG deep learning model for heart failure performed significantly worse in young Black patients—especially Black women—than in other groups, illustrating how algorithmic bias may aggravate health outcome disparities (Kaur et al, 2024). If left unaddressed, such performance gaps could exacerbate healthcare inequities by offering less accurate or delayed diagnoses to minority populations (Mihan et al, 2024). To promote fairness and generalizability, it is crucial to train and validate AI-ECG models on diverse, multi-ethnic datasets and to report model performance stratified by race, ethnicity, age, and sex (Noseworthy et al, 2020). Together, these considerations point out that ensuring fairness and generalizability across diverse demographic groups is

not only a technical requirement but a crucial step toward the equitable clinical translation of AI-ECG.

# Conclusion

In conclusion, the integration of AI into ECG analysis represents a paradigm shift across the entire spectrum of cardiovascular diagnostics and management. AI models have shown remarkable accuracy in both traditional and emerging ECG applications, extending this century-old tool into broader clinical use. This advancement not only enhances the detection of various cardiac conditions but also supports the development of more personalized and timely interventions, ultimately improving patient outcomes. Prospective studies and RCTs have demonstrated the potential of AI-ECG systems to improve patient outcomes, including reduced mortality and shorter door-to-balloon times in acute settings. Moreover, the concept of opportunistic screening through AI-ECG offers a promising avenue for early detection of asymptomatic cardiac conditions during routine assessments. However, challenges remain, including the need for further research to refine these models, enhance interpretability, and ensure seamless integration into clinical workflows. Addressing these challenges will be crucial to fully realize the transformative potential of AI-ECG in enhancing diagnostic accuracy, guiding timely interventions, and ultimately improving cardiovascular health outcomes.

## Pending issues

Despite remarkable advances in AI-enabled ECG, several important challenges remain that warrant further investigation. First, current models are often trained on single-center or homogeneous datasets. Broader validation across diverse populations, devices, and healthcare systems is essential to ensure equitable performance and avoid algorithmic bias. Second, there is no consensus on acquisition protocols, alert thresholds, or interpretability requirements. Establishing operational frameworks will be pivotal for safe and reproducible deployment in routine care. Third, defining acceptable alert rates, escalation pathways, and strategies to mitigate clinician alert fatigue are necessary to facilitate real-world adoption. Fourth, AI-ECG models remain largely "black boxes." Improved explainability may enhance clinician trust, reveal novel disease biology, and stimulate new avenues for translational research. Finally, while early randomized trials demonstrate reduced mortality and expedited treatment times, further large-scale prospective studies are needed to confirm durability of benefit, cost-effectiveness, and impact on healthcare systems.

# Peer review information

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

## Acknowledgements

This study was supported by funding from the National Science and Technology Council, Taiwan (NSTC 114-2321-B-016-005 to S-H Lin).

## Author contributions

**Chin-Sheng Lin**: Writing—original draft; Writing—review and editing. **Wei-Ting Liu**: Writing—review and editing. **Yuan-Hao Chen**: Writing—review and editing. **Shih-Hua Lin**: Writing—review and editing. **Chin Lin**: Investigation; Writing—original draft; Writing—review and editing.

## Disclosure and competing interests statement
The authors declare no competing interests.

