## [Peer Review File · EMBO Molecular Medicine]

Artificial Intelligence-enabled Electrocardiography from Scientific Research to Clinical Application

Chin-Sheng Lin, Wei-Ting Liu, Yuan-Hao Chen, Shih-Hua Lin, and Chin Lin

Corresponding author: Chin Lin (xup6fup@mail.ndmctsgh.edu.tw)

Review Timeline:

Submission Date:	2nd Jul 25
Editorial Decision:	28th Aug 25
Revision Received:	6th Oct 25
Editorial Decision:	24th Oct 25
Revision Received:	27th Oct 25
Accepted:	4th Nov 25

Editor: Zeljko Durdevic

Transaction Report:

28th Aug 2025

Dear Dr. Lin,

Thank you for the submission of your manuscript to EMBO Molecular Medicine and please accept my apologies for the delay in getting back to you due to the holiday season. We have now received feedback from the two reviewers who agreed to evaluate your manuscript. As you will see from the reports below, the referee #1 is critical about the lack of detail regarding the latest development of AI-enabled ECG, while the referee #2 recognizes the interest and timeliness of the review but also raises serious concerns particularly regarding the limited clinical impact. Therefore, we would like to invite major revision of the manuscript with the understanding that all referee concerns should be addressed, and all referee suggestions implemented.

I would also like to ask you to amend the following:

- 1) Figures 1,2 and 4 seem somewhat redundant and should be combined into one simplified figure with 2-3 panels presenting paradigm shift from conventional ECG to AI driven approach.
- 2) Remove all figures from the manuscript and leave only their legends on the end of the main manuscript file.
- 3) Place all tables at the end of the manuscript file after figure legends.
- 4) Rename "Funding" to "Acknowledgements".
- 5) Add "Disclosure and competing interests statement". We updated our journal's competing interests policy in January 2022 and request authors to consider both actual and perceived competing interests. Please review the policy <https://www.embopress.org/competing-interests> and update your competing interests if necessary.
- 6) Glossary: The glossary is meant to explain some of the terms used for laymen. Could you please identify terms that may need an "explanation"?
- 7) Pending issues: At the end of each article is a box highlighting issues that still need further studies and where research efforts should converge. Could you identify some pending issues?
- 8) If BioRender was used to create the figures, please add following sentence to the figure legends: "Graphics were created with BioRender.com.
- 9) Please correct the reference citation in the text and reference list. In the text a reference should be cited by author and year of publication. Include a space between a word and the opening parenthesis of the reference that follows. In the reference list, citations should be listed in alphabetical order. Where there are more than 10 authors on a paper, 10 will be listed, followed by "et al.". Also, please remove DOIs. DOIs should only be used for preprints and datasets that have not been published. Please check "Author Guidelines" for more information. <https://www.embopress.org/page/journal/17574684/authorguide#referencesformat>

Further consideration of a revision that addresses reviewer's concerns in full will entail an additional round of review. Acceptance or rejection of the manuscript will depend on the completeness of your responses included in the next, final version of the manuscript. For this reason, and to save you from any frustrations in the end, I would strongly advise against returning an incomplete revision.

I hope that the referees' comments do not prove too problematic to address and I look forward to reading your next version.

Yours sincerely,

Zeljko Durdevic

Zeljko Durdevic
Senior Editor
EMBO Molecular Medicine

*** IMPORTANT INFORMATION ***

- 1) a .doc formatted version of the manuscript text (including Figure legends and tables)
- 2) Separate figure files
- 3) a letter INCLUDING the reviewer's reports and your detailed responses to their comments.

Also, and to save some time should your paper be accepted, please read below for additional information regarding some features of our research articles:

1) Glossary: EMBO Molecular Medicine articles will be accompanied by a glossary explaining some of the terms used for laymen. I identified the following:

_____, _____, _____

Could you please help us in identifying terms that may need an "explanation" other terms that we can add to the glossary.

2) For more information: This is a short list of related web links for further consultation by the readers. Could you identify some relevant ones? Examples are patient associations, OMIM related links, databases, authors websites, etc.

3) Pending issues: At the end of each article we will have a box highlighting issues that still need further studies and where research efforts should converge (we call this the Pending issues box). From my reading I would say:

but I can see there may be many more. Could you work on this as well?

4) Disclosure and competing interest statement: Please include a statement declaring any competing commercial interests in relation to your submitted work.

5) Please note that we now mandate that all corresponding authors list an ORCID digital identifier. This takes <90 seconds to complete. We encourage all authors to supply an ORCID identifier, which will be linked to their name for unambiguous name identification.

Currently, our records indicate that the ORCID for your account is 0000-0003-2337-2096.

Link Not Available

-

Thank you,

Zeljko Durdevic

Zeljko Durdevic
Senior Editor
EMBO Molecular Medicine

***** Reviewer's comments *****

Referee #1 (Remarks for Author):

In this paper, the authors presented a brief overview on the potential application of AI. While this review cover some important general concepts after the application of AI in the ECG analysis, there is lack of detail or summary in term of the latest development on the analysis of different features beyond conventional morphological analysis of ECG using AI. Moreover, it will be useful for the readers to have a better overview on the applications in the diagnosis of STEMI, heart failure and cardiomyopathy in form a Table with all the recent data. Similarly, a Table should be include to provide a summary on the ECG AI to predict diseases eg sudden death and atrial fibrillation.

Referee #2 (Remarks for Author):

The manuscript presents a timely and translationally oriented review that captures the paradigm shift of electrocardiography from rule-based interpretation to AI-driven discovery and patient benefit. Its novelty lies in highlighting RCT evidence, outcome-based practice changes, and emerging frameworks such as OMI detection and "previvor" management. Though this is a well-structured review with evidence that AI-ECG detects conditions invisible to clinicians, such as low EF, silent AF, and risk classification, there are still many questions needed to be addressed to improve the scholarly and clinical impact.

Major comments:

1. Expand discussion on dataset shift and demographic performance to address issues of generalizability and fairness.
2. To propose operational frameworks for deployment, including alert thresholds, acquisition standards, and interpretability protocols.
3. To add operational guidance for PPV/alert-rate targets, escalation pathways, and example triage protocols.
4. Clarify clinical ramifications and provide head-to-head prospective studies comparing AI-OMI triage vs STEMI criteria.
5. Add a PRISMA-style evidence map of RCTs/prospective studies.
6. Extend Table 1 with calibration and deployment metrics (PPV/NPV, alert rate, workflow latency).

Response to Reviewer #1:**General comments:**

1. *In this paper, the authors presented a brief overview on the potential application of AI. While this review cover some important general concepts after the application of AI in the ECG analysis, there is lack of detail or summary in term of the latest development on the analysis of different features beyond conventional morphological analysis of ECG using AI.*

Reply:

We thank the reviewer for this insightful suggestion. In the revised manuscript, we added a new subsection "Feature representations in AI-ECG beyond conventional morphology" to summarize the advantage of modern deep learning models compared to the previous feature-based machine learning models.

2. *Moreover, it will be useful for the readers to have a better overview on the applications in the diagnosis of STEMI, heart failure and cardiomyopathy in form a Table with all the recent data.*

Reply:

We agree and have added Table 2 summarizing recent prospective diagnostic studies in left ventricular dysfunction (heart failure), paroxysmal atrial fibrillation, hypertrophic cardiomyopathy, and STEMI (we excluded STEMI only defined by ECG because it is a traditional application) including study population and key performance metrics (AUROC/SENS/SPEC/PPV/NPV when reported).

3. *Similarly, a Table should be include to provide a summary on the ECG AI to predict diseases eg sudden death and atrial fibrillation.*

Reply:

Thank you for this helpful request. We have added Table 3 to summarize key prognostic AI-ECG models, including atrial fibrillation (incident AF) and mortality risk prediction. As described in the revised text, we highlight the consistent performance degradation observed during external validation (typically a 5–20% drop in AUROC), as well as the challenges in applying internally derived cutoffs across populations. Despite these limitations, most models retained AUROC values above 0.65, supporting the biological plausibility of AI-ECG in predicting long-term outcomes.

Response to Reviewer #2:

General comments:

1. *The manuscript presents a timely and translationally oriented review that captures the paradigm shift of electrocardiography from rule-based interpretation to AI-driven discovery and patient benefit. Its novelty lies in highlighting RCT evidence, outcome-based practice changes, and emerging frameworks such as OMI detection and "previvor" management.*

Reply:

We thank the reviewer for this encouraging assessment.

2. *Though this is a well-structured review with evidence that AI-ECG detects conditions invisible to clinicians, such as low EF, silent AF, and risk classification, there are still many questions needed to be addressed to improve the scholarly and clinical impact.*

Reply:

We thank the reviewer for this constructive guidance. In response, we added a targeted implementation-focused package to strengthen both scholarly and clinical impact as below response to specific comments. Together, these additions directly address your concern by turning conceptual advances into operational guidance that clinicians and program leads can adopt.

Specific comments:

1. *Expand discussion on dataset shift and demographic performance to address issues of generalizability and fairness.*

Reply:

Thank you for your comment. We have added a paragraph starting in "Finally, many AI-ECG models are developed using training data predominantly from a single ethnic group or geographic region. ..." in the end of the section "Future directions and recommendations" discussing generalizability and fairness.

2. *To propose operational frameworks for deployment, including alert thresholds, acquisition standards, and interpretability protocols. To add operational guidance for PPV/alert-rate targets, escalation pathways, and example triage protocols.*

Reply:

Thank you for your comment. We have added a section "Frameworks for AI-ECG deployment" discussing operational frameworks for deployment, as well as operational guidance, to better elucidate the concept.

3. *Clarify clinical ramifications and provide head-to-head prospective studies comparing AI-OMI triage vs*

STEMI criteria.

Reply:

We thank the reviewer for this important comment. We have now incorporated a head-to-head study (Herman et al., 2024) directly comparing AI-based OMI triage, conventional STEMI criteria, and expert interpretation. As highlighted in the revised text, this study demonstrated that AI achieved markedly higher sensitivity (0.806) while maintaining high specificity (0.937), compared with STEMI criteria (sensitivity 0.325; specificity 0.977) and expert cardiologists (sensitivity 0.730; specificity 0.957). These findings underscore the superior diagnostic performance of AI-ECG in detecting true coronary occlusion and support the clinical relevance of AI-OMI triage as a transformative approach in acute myocardial infarction management.

4. Add a PRISMA-style evidence map of RCTs/prospective studies.

Reply:

We thank the reviewer for this suggestion. In the revised manuscript, we have added a PRISMA 2020 flow diagram (Supplemental Figure 1) in Supplementary Methods summarizing the identification, screening, eligibility, and inclusion of studies, with explicit counts for randomized controlled trials (RCTs) and prospective clinical trials. The diagram also lists reasons for full-text exclusion (e.g., not AI-ECG, not RCT/prospective). To ensure transparency, the Supplementary Methods specifies the sources searched (PubMed/MEDLINE), date of last search, and all exclusion categories. This addition provides the requested evidence map and clarifies how the included RCTs/prospective studies were selected. In conclusion, we added Table 4 to summarize 11 included studies, and revised the descriptions in "AI-ECG in diagnostic support and opportunistic screening" and "AI-ECG in community screening" sections.

5. Extend Table 1 with calibration and deployment metrics (PPV/NPV, alert rate, workflow latency).

Reply:

We agree that deployment-facing metrics are essential for clinical translation. However, Table 1 summarizes results from human–AI head-to-head challenges with case-enrichment and non-clinical disease prevalence, which makes PPV/NPV non-portable and potentially misleading outside the challenge context. To avoid conflating settings, we retained Table 1 as a threshold-independent comparison (e.g., AUROC/AUPRC, accuracy, sensitivity/specificity when applicable) and added a footnote explicitly stating that PPV/NPV are not reported due to the enriched prevalence and contest design. Moreover, we added two new tables (Tables 2 and 3 for diagnostic and prognostic deployment) that supply the requested deployment metrics under real-world prevalence.

24th Oct 2025

Dear Dr. Lin,

Thank you for the submission of your revised manuscript to EMBO Molecular Medicine. Below you will find the final referee report. I am pleased to inform you that we will be able to accept your manuscript pending the following final amendments:

- 1) Add up to 5 keywords.
- 2) Figures:
 - We have sent the figures 1 and 2 for the final adjustments to our graphic designer, who will contact you for your approval.
 - Figure 3 seems to show real data. Is that the case? Would it be possible to replace it with schematics? The figure in its current form is difficult to understand for a nonspecialist reader. Please replace lowercase a and b with uppercase A and B.
- 3) Please remove "For more information".
- 4) Rename "Supplementary Methods" to "Appendix" and add table of content on the first page with page numbers. Rename Supplemental Figure 1 to Appendix Figure S1 and Supplementary References to References.
- 5) As part of the EMBO Publications transparent editorial process EMBO Molecular Medicine will publish online a Review Process File (RPF) to accompany accepted manuscripts. This file will be published in conjunction with your paper and will include the anonymous referee reports, your point-by-point response and all pertinent correspondence relating to the manuscript. Let us know whether you agree with the publication of the RPF.

I look forward to receiving the revised version of your manuscript.

Yours sincerely,

Zeljko Durdevic

Zeljko Durdevic
Senior Editor
EMBO Molecular Medicine

*** IMPORTANT INFORMATION ***

- 1) a .doc formatted version of the manuscript text (including Figure legends and tables)
- 2) Separate figure files
- 3) a letter INCLUDING the reviewer's reports and your detailed responses to their comments.

Also, and to save some time should your paper be accepted, please read below for additional information regarding some features of our research articles:

- 1) Glossary: EMBO Molecular Medicine articles will be accompanied by a glossary explaining some of the terms used for laymen. I identified the following:

____, _____, _____

Could you please help us in identifying terms that may need an "explanation" other terms that we can add to the glossary.

2) Pending issues: At the end of each article we will have a box highlighting issues that still need further studies and where research efforts should converge (we call this the Pending issues box). From my reading I would say:

but I can see there may be many more. Could you work on this as well?

3) Disclosure and competing interest statement: Please include a statement declaring any competing commercial interests in relation to your submitted work.

4) Please note that we now mandate that all corresponding authors list an ORCID digital identifier. This takes <90 seconds to complete. We encourage all authors to supply an ORCID identifier, which will be linked to their name for unambiguous name identification.

Currently, our records indicate that the ORCID for your account is 0000-0003-2337-2096.

Link Not Available

-

Thank you,

Zeljko Durdevic

***** Reviewer's comments *****

Referee #2 (Remarks for Author):

The revised manuscript substantially enhances academic rigor, clinical relevance, and translational value. A new subsection on "Feature representations in AI-ECG beyond conventional morphology" clarifies how deep learning surpasses traditional methods, enriching conceptual depth. Incorporating a PRISMA-guided evidence map and new summary tables (Tables 2-4) strengthens methodological transparency and highlights AI-ECG's diagnostic and prognostic potential. The added section, "Frameworks for AI-ECG deployment," provides practical implementation guidance, emphasizing interpretability and fairness. Collectively, these revisions transform the manuscript into a comprehensive, evidence-based, and globally relevant reference that integrates algorithmic innovation with clinical application, now presented in an academically acceptable format.

The authors addressed the remaining editorial issues.

4th Nov 2025

Dear Dr. Lin,

We are pleased to inform you that your manuscript is accepted for publication and is now being sent to our publisher to be included in the next available issue of EMBO Molecular Medicine pending the final figure adjustments by our graphic designer, who will contact you for your approval of the figures.

Your manuscript will be processed for publication by EMBO Press. It will be copy edited and you will receive page proofs prior to publication. You will soon be contacted by Springer Nature to sign your publishing license. When you login to the customer service website, please use the following token to waive the article publication charges. Should you experience any difficulty, please email publishing@embo.org.

Waiver token: [removed]

Zeljko Durdevic
Senior Editor
EMBO Molecular Medicine